# WavePaint: Resource-efficient Token-mixer for Self-supervised Inpainting

## Abstract

Inpainting, which refers to the synthesis of missing regions, can help restore occluded or degraded areas of an image and also serve as a precursor task for self-supervision of neural networks for computer vision. The current state-of-the-art models for inpainting are computationally heavy as they are based on transformer or CNN backbones that are trained in adversarial or diffusion settings. This paper diverges from vision transformers by using a computationally-efficient WaveMix-based fully convolutional architecture—WavePaint. It uses a 2D-discrete wavelet transform (DWT) for spatial and multi-resolution token-mixing along with convolutional layers. The proposed model outperforms the current state-of-the-art models for image inpainting on reconstruction quality while also using much fewer parameters and GPU RAM, and considerably lower training and evaluation times. Our model even outperforms current GAN-based architectures in CelebA-HQ dataset without using an adversarially trainable discriminator. This work suggests that neural architectures that are modeled after natural image priors require fewer parameters and computations to achieve better generalization.

## 1 Introduction

Image inpainting refers to the process of filling missing parts of an image (blemishes, holes, and other defects) realistically to match the available context, thereby restoring the degraded image. It requires implicitly modeling large scale structures in natural images and an ability to perform image synthesis. State-of-the-art inpainting models are based on deep neural networks (Li et al., 2020; Liu et al., 2018) trained in a self-supervised and adversarial manner (Pathak et al., 2016; Ma et al., 2019; Yu et al., 2018; 2019) by automatically generating training samples from large image datasets by randomly masking parts of the image.

Specifically, we have worked on large mask inpainting, where the mask occludes a substantial and non-trivial part of the image. Such image reconstruction tasks require networks to have large effective receptive

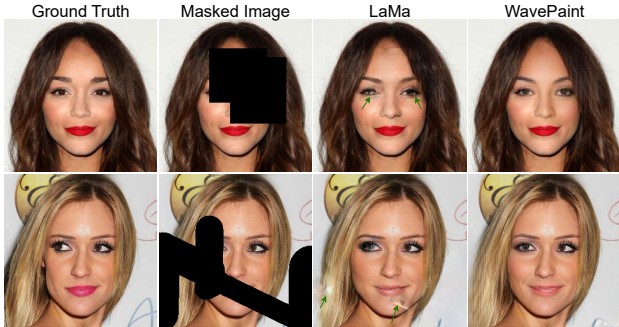

Figure 1: Qualitative comparison of inpainted images generated by WavePaint and LaMa (Suvorov et al., 2021a). The green arrows point to improper image completion.

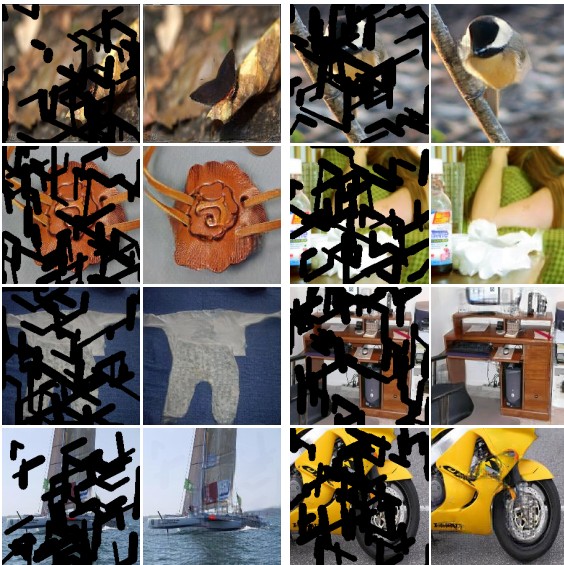

Figure 2: Masked images from the ImageNet validation set (first and third columns) and their inpainted versions (second and fourth columns) using WavePaint

fields (Luo et al., 2017). Convolutional neural networks (CNN) can generate visually plausible image structures and textures, but adds distorted or blurry features inconsistent with surrounding areas due to its ineffectiveness in explicitly borrowing information from distant spatial locations (Yu et al., 2018). CNNs also require deep architectures (a large number of layers) with residual connections for increasing their receptive fields (He et al., 2015). On the other hand, using self-attention to access all the pixels of an image right from the first layer gives transformers large receptive fields. However, their quadratic complexity with respect to sequence length (number of patches) introduces an enormous computational burden. Moreover, transformers require larger training data than CNNs, since they lack the inductive bias of spatial equivariance (Khan et al., 2022).

The search for efficient models that can mix global spatial information while retaining the inductive bias of CNNs has led to the development of token-mixing models such as PoolFormer (Yu et al., 2022), ConvMixer (Trockman & Kolter, 2022) and WaveMix (Jeevan et al., 2023) which use pooling, depth-wise convolutions and 2 dimensional-discrete wavelet transform (2D-DWT), respectively. These alternatives consume only a fraction of the resources compared to transformers to achieve competitive generalization in tasks such as classification and segmentation. The performance of these models on image generation or restoration tasks has not been evaluated.

Our model is a neural architecture that is inspired by WaveMix (Jeevan et al., 2022; 2023) and ConvMixer (Trockman & Kolter, 2022). We investigated the application of WaveMix architectural framework to the task of image inpainting with suitable adaptations. This choice is motivated by the state-of-the-art performance of WaveMix in multiple datasets on the task of parameter-efficient image classification and segmentation by modeling additional inductive priors of images, such as scale invariance, and also due to its exponential receptive field expansion.

We have not worked on blind mask inpainting, where the model does not see the mask. Sending mask to the model is necessary in the large-mask setting for the model to know where the mask is and where to fill information. None of models discussed or compared in this paper use blind mask inpainting.

Our contributions are summarized below:

- We present—WavePaint—a token-mixing network modeled after natural image priors that can perform image inpainting. The network is based on recently proposed WaveMix architecture which uses 2D-DWT for spatial token-mixing (Jeevan et al., 2022; 2023). We also employ depth-wise convolu-

tion in our network for additional token-mixing. The presence of wavelet layers enables the model to have faster receptive field expansion compared to CNNs, which helps in better image reconstruction through access to global context of the image. Figure 9 shows how certain macro artifacts are reduced by using WavePaint compared with LaMa (Suvorov et al., 2021a), for instance. Also, see sample results on ImageNet validation set in Figure 2.

- The use of a paramter-free 2D-DWT and parameter-efficient depth-wise convolution helps WavePaint reconstruct images without the need for a large number of parameters. WavePaint with just 5M parameters can outperform much larger models such as LaMa (27 M) (Suvorov et al., 2021a) and CoModGAN (109 M) (Zhao et al., 2021) on CelebA-HQ dataset for multiple mask sizes. WavePaint also consumes fewer resources and less time for training and inference.

- WavePaint does not need adversarial or diffusion-based training routines, which are slow. The ability of wavelet token mixing to generate realistic images from masked ones shows that we can develop more efficient neural networks for image generation in particular and image processing in general.

- Our model reconstructs the image using a simple single-stage network, as opposed to the complicated multi-stage models that have been proposed to generate intermediate predictions to restore the missing parts (Liu et al., 2020; Nazeri et al., 2019b; Song et al., 2018).

- We show that utilizing natural image priors in neural architectural design may be the way forward to avoid large computational costs and training datasets.

## 2    Related Works

Early approaches to image inpainting relied on propagating appearance information from neighbouring pixels to masked area (Ballester et al., 2001; Bertalmío et al., 2000) such as patch-based methods which used similar patches to complete missing areas. Data driven deep learning methods started exploiting encoder-decoder architecture (Yeh et al., 2017; Nazeri et al., 2019a; Zhu et al., 2021), assuming that masked image, once encoded, will have adequate information for reconstruction. Residual connections (He et al., 2015) and multi-branch convolutions with reduced kernels (Szegedy et al., 2014) enabled the use of deeper CNN models. Mask-Aware Dynamic Filtering (MADF) (Zhu et al., 2021) uses an encoder-decoder framework to learn multi-scale features for missing regions in the encoding phase. Such methods could only handle narrow masks with small color and texture variation.

Adversarial training is the most prevalent approach for inpainiting today (Zhao et al., 2021; Li et al., 2022; Suvorov et al., 2021a). Co-ModGAN (Zhao et al., 2021) introduces variability into the generated outputs by integrating input image-conditional and unconditional generators. Image completion with transformer (ICT) (Wan et al., 2021) is a transformer-CNN hybrid that uses transformers to model the long-range relationships in images to recover pluralistic coherent structures together with coarse textures, and uses CNN for texture replenishment. Mask-Aware Transformer (Li et al., 2022) uses a multi-head contextual attention for long-range dependency modeling by exploiting valid tokens indicated by a dynamic mask for directly processing high-resolution images. It also proposed a modified transformer block to increase the stability of large mask training. While attention-based models (Li et al., 2022; Wan et al., 2021) enable global token-mixing better than CNN based models do, their quadratic complexity of attention requires large computational resources and data sizes.

Alternatives to pure transformers for inpainting have attempted to use image transforms that mix spatial tokens. For instance, 2D-DWT was used by WaveFill (Yu et al., 2021) to decompose images into multiple frequency bands and fill the missing regions in each frequency band separately. LaMa (Suvorov et al., 2021a) used Fourier convolution blocks having image-wide receptive field and showed that larger receptive fields aid large mask inpainting. They also used an aggressive large mask generation strategy for better generalization on different mask sizes. LaMa was shown to be more efficient than other models trained on adversarial settings. We employ the same large mask generation scheme used by LaMa for our experiments.

Adversarial training is widely used for training inpainting models due to its ability to generate visually plausible details. It requires the use of separate discriminator model whose parameters are updated during

training, which requires more memory and time. Generative adversarial networks (GAN) also suffer from mode collapse, training instabilities, and require extensive hyper-parameter tuning (Zhang et al., 2019; 2018b; Kodali et al., 2017).

Recently, diffusion models have emerged as an alternative to GANs (Suvorov et al., 2021b; Lugmayr et al., 2022). Diffusion models uses a T-fold pass through a fixed network to go from completely random noise to a coherent and contextually consistent image. Latent diffusion model (LDM) (Rombach et al., 2021) works on a lower-dimensional feature space rather than the image space to address the time-consuming nature of diffusion training. The model uses an encoder-decoder architecture with the slow diffusion step at the neck of the chain to speed up the entire network. RePaint (Lugmayr et al., 2022) is a denoising diffusion probabilistic model (DDPM) based inpainting approach which employs a pretrained unconditional DDPM as the generative prior. It only alters the reverse diffusion iterations by sampling the unmasked area to condition the generation process.

Even though diffusion models can generate realistic images, the training and inference processes require large computational resources. It also requires careful tuning of hyper-parameters which can also be time-consuming. Additionally, generating images with diffusion models requires multiple iterations, resulting in much longer inference times compared to other methods.

In this work, we develop a novel framework that simultaneously achieves high quality image generation and large mask inpainting, without using adversarial or diffusion training, thereby reducing the demand for compute and memory. To achieve our goals, we developed a novel neural network architecture that is based on natural image priors of translational equivariance, scale invariance, and feature locality, inspired from WaveMix (Jeevan et al., 2023).

## 3 WavePaint Architectural Framework

Inspired by the success of WaveMix (Jeevan et al., 2023) and ConvMixer (Trockman & Kolter, 2022), which use 2D-DWT and depthwise-convolutions, respectively, for parameter efficient token-mixing, we propose a neural architecture that can inpaint masked images using these token-mixing operations. The ability of these token-mixers to impart rapid receptive field expansion from initial layers itself helps the model grasp the global context faster than conventional CNN-based networks. Unlike other popular models for image inpainting that uses diffusion or adversarial training, our model has simple single network architecture and can perform well without the need for a discriminator network.

### 3.1 Overall architecture

The input image $\mathbf{x} \in \mathbb{R}^{H \times W \times 3}$ is masked by a binary mask $m \in \{0, 1\}^{H \times W \times 1}$ that is generated from a mask generator. The masked image is denoted as $\mathbf{x} \oplus m$. The mask $m$ is concatenated with the masked image $\mathbf{x} \oplus m$, resulting in a 4-channel input $\hat{\mathbf{x}} \in \mathbb{R}^{H \times W \times 4}$ that is passed to the model as shown in Figure 3.

The network consists of a series of $M$ Wave modules which processes the input $\hat{\mathbf{x}}$ and gives the output $\hat{\mathbf{y}} \in \mathbb{R}^{H \times W \times 3}$, which is multiplied by the inverted binary mask $1 - m$ to hide the unmasked areas of the output and retains the inpainted parts by the model. This is added back to the masked image $\hat{\mathbf{x}}$ which fills the unmasked areas and creates the final inpainted image $\mathbf{y} \in \mathbb{R}^{H \times W \times 3}$. This ensures that the model only fills the masks areas and not change pixel information of unmasked parts.

### 3.2 Wave Modules

Proper inpainting requires global context information of the image. WaveMix has shown rapid expansion of receptive fields from very early layers (Jeevan et al., 2023). So we use 4 WaveMix blocks in series in each of the Wave modules to process the image and get global context. This is further aided by the depth-wise convolution layer which further helps with spatial token-mixing with high parameter-efficiency (Trockman & Kolter, 2022).

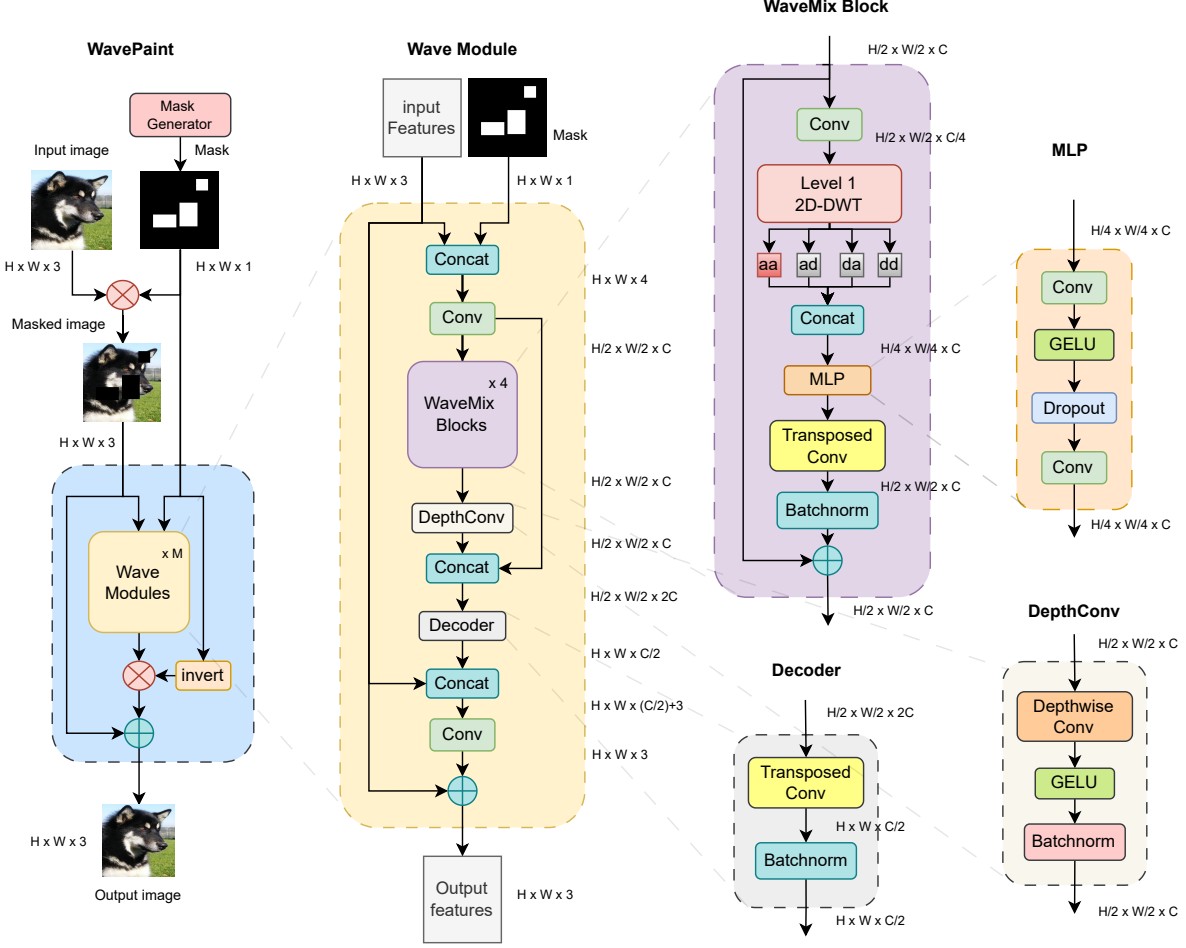

Figure 3: Architecture of WavePaint along with details of Wave module, WaveMix block (Jeevan et al., 2023), Decoder, DepthConv and MLP along with the resolutions of feature maps shown after each operation

Denoting input and output tensors of the Wave module by $\hat{\mathbf{x}}_{in}$ and $\hat{\mathbf{x}}_{out}$, respectively; convolution operations by $c_1$ and $c_2$ and its respective trainable parameter sets by $\theta_1$ and $\theta_2$ respectively; the series of WaveMix blocks by $WB$; DepthConv by $DC$; Decoder by $D$; concatenation along the channel dimension by $\oplus$, and point-wise addition by $+$, the operations inside a Wave module can be expressed using the following equations:

$$\hat{\mathbf{x}}_0 = \hat{\mathbf{x}}_{in} \oplus m; \qquad \hat{\mathbf{x}}_{in} \in \mathbb{R}^{H \times W \times 4} \qquad (1)$$

$$\hat{\mathbf{x}}_1 = c_1(\hat{\mathbf{x}}_0, \theta_1); \qquad \hat{\mathbf{x}}_1 \in \mathbb{R}^{H/2 \times W/2 \times C} \qquad (2)$$

$$\hat{\mathbf{x}}_2 = WB(\hat{\mathbf{x}}_1); \qquad \hat{\mathbf{x}}_2 \in \mathbb{R}^{H/2 \times W/2 \times C} \qquad (3)$$

$$\hat{\mathbf{x}}_3 = DC(\hat{\mathbf{x}}_2); \qquad \hat{\mathbf{x}}_3 \in \mathbb{R}^{H/2 \times W/2 \times C} \qquad (4)$$

$$\hat{\mathbf{x}}_4 = \hat{\mathbf{x}}_3 \oplus \hat{\mathbf{x}}_1; \qquad \hat{\mathbf{x}}_4 \in \mathbb{R}^{H/2 \times W/2 \times 2C} \qquad (5)$$

$$\hat{\mathbf{x}}_5 = D(\hat{\mathbf{x}}_4); \qquad \hat{\mathbf{x}}_5 \in \mathbb{R}^{H \times W \times C/2} \qquad (6)$$

$$\hat{\mathbf{x}}_6 = \hat{\mathbf{x}}_5 \oplus \hat{\mathbf{x}}_{in}; \qquad \hat{\mathbf{x}}_6 \in \mathbb{R}^{H \times W \times (C/2+3)} \qquad (7)$$

$$\hat{\mathbf{x}}_7 = c_2(\hat{\mathbf{x}}_6, \theta_2); \qquad \hat{\mathbf{x}}_7 \in \mathbb{R}^{H \times W \times 3} \qquad (8)$$

$$\hat{\mathbf{x}}_{out} = \hat{\mathbf{x}}_7 + \hat{\mathbf{x}}_{in}; \qquad \hat{\mathbf{x}}_{out} \in \mathbb{R}^{H \times W \times 3} \qquad (9)$$

Each Wave module receives the input $\hat{\mathbf{x}}_{in} \in \mathbb{R}^{H \times W \times 3}$ and the mask $m$ which are concatenated to create $\hat{\mathbf{x}}_0$ (Eq. 11). $\hat{\mathbf{x}}_0$ is send to a convolution layer $c_1$ that reduces its feature resolution by half and increases the channel dimension to $C$ (Eq. 12). This feature map $\hat{\mathbf{x}}_1$ is sent to a series of four WaveMix blocks for token-mixing (Eq. 13). The output from the WaveMix block $\hat{\mathbf{x}}_2$ is further passed through a DepthConv module where the feature maps undergo further spatial token-mixing from the depth-wise convolution (Eq. 14). A skip connection from $c_1$ is concatenated with the output from DepthConv module $\hat{\mathbf{x}}_3$ which increases the channel dimension of the output $\hat{\mathbf{x}}_4$ to $2C$ (Eq. 15). This output is further passed through a Decoder network which increases the resolution of feature maps to original resolution (Eq. 6). The Decoder layer also reduces the number of channels to $C/2$ and the feature maps $\hat{\mathbf{x}}_5$ are again concatenated with the input $\hat{\mathbf{x}}_{in}$ (Eq. 7). The output after concatenation $\hat{\mathbf{x}}_6$ is then passed to a final convolution layer $c_2$ to generate the output $\hat{\mathbf{x}}_7$ (Eq. 8). A residual connection (He et al., 2015) is also provided from the input for ease of gradient flow (Eq. 9) and resultant feature maps are the final output of the Wave module $\hat{\mathbf{x}}_{out}$.

### 3.3 WaveMix Blocks

The WaveMix block (Jeevan et al., 2023) is the fundamental building block of WaveMix architecture which allows multi-resolution token-mixing of information using 2D-DWT. This helps in a rapid expansion of the receptive field with depth. It also reduces the computational burden because 2D-DWT decreases the input resolution by half and further processing by multi-layer perceptron (MLP) is faster and cheaper. DWT helps in lowering the number of model parameters significantly, as it lacks any parameters, while promoting global context understanding even in a shallow network. We have used the WaveMix block with one level of 2D-DWT using Haar wavelet due to its simplicity, speed and higher performance (when used in WaveMix compared to other wavelets (Jeevan et al., 2023)). Details of the operations inside WaveMix block are provided in Appendix.

### 3.4 DepthConv

The DepthConv module utilizes a depthwise convolution operation followed by a GELU activation and batch normalization, as depicted in Figure 3. We employ a depthwise convolution with a kernel size of 5, which is smaller than that used in ConvMixer models. This design choice reduces the parameter count while maintaining effective spatial token mixing. Depthwise convolution acts as an additional token-mixing mechanism that complements the wavelet-based token-mixing process. While the wavelet-based token mixing is a parameter-free operation, the primary learning occurs through parameters in the MLPs and transposed convolution blocks that follow the wavelet transforms in each layer. This architecture necessitates adding more WaveMix layers to enhance expressiveness. By integrating depthwise convolution, the network's expressiveness is improved with a minimal increase in parameter count, leveraging the efficiency of depthwise convolution for parameter-efficient design.

### 3.5 Decoder

Decoder module is used to up-sample the resolution of feature maps back to original input resolution to the Wave module. It comprises of a transposed convolution layer followed by batch-normalization as shown in Figure 3. The transposed convolution layer is also used to reduce the number of channels by 4, from $2C$ to $C/2$.

## 4 Experiments and Results

In this section, we report the datasets and metrics used for extensive comparison with the state-of-the-art, the loss functions and other implementation details, and quantitative image reconstruction accuracy comparison, along with compute time, and memory requirements comparison. We also show a sample of qualitative results. Additionally, we report ablation studies for various architectural choices.

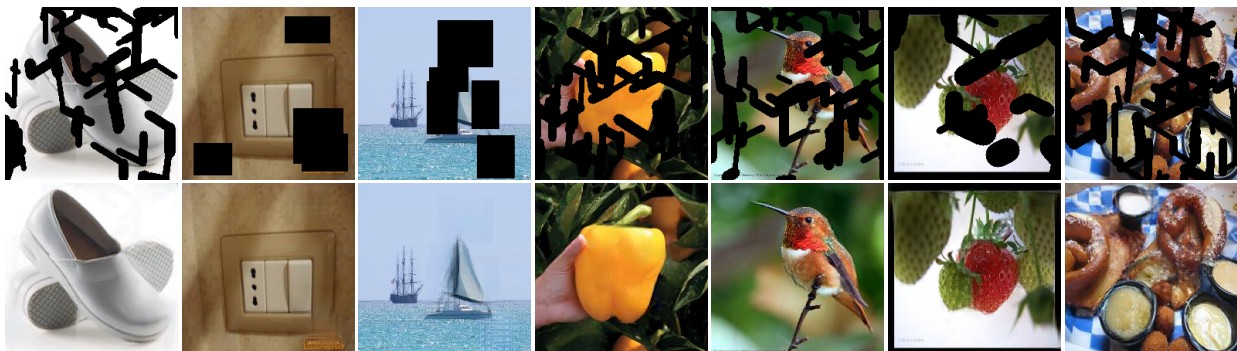

Figure 4: Masked images from the ImageNet validation set (top row) and their inpainted versions generated by WavePaint (bottom row).

### 4.1 Datasets and Metrics

We use CelebA-HQ (Karras et al., 2018), Places-365 standard (Zhou et al., 2017) and ImageNet (Deng et al., 2009) datasets for our experiments. We use images of size $256 \times 256$ and $512 \times 512$ for CelebA-HQ, $256 \times 256$ for Places-365 and $224 \times 224$ for ImageNet experiments. We followed the same mask generation policy employed in LaMa (Suvorov et al., 2021a) and used their settings to generate narrow, medium and wide masks. We only used the wide mask for training and evaluated the models on narrow, medium and wide masks. We took 26,000 images for training and 2,000 for testing from $256 \times 256$ CelebA-HQ dataset similar to the procedure followed by (Suvorov et al., 2021a) and 24,182 images for training and 2993 for testing from $512 \times 512$ CelebA-HQ dataset as used by (Li et al., 2022). For ImageNet and Places-365 dataset, we used the training set for training and validation set for inference.

Learned perceptual image patch similarity (LPIPS) (Zhang et al., 2018a) and Fréchet inception distance (FID) (Heusel et al., 2018) are reported as metrics since L1 and L2 distances are not enough to compare inpainted images with large masks where multiple natural completions are possible. Inference and training throughput on a single GPU was reported in frames/sec (FPS), i.e., images per second.

### 4.2 Loss function and implementation details

We used a hybrid loss $L_{hybrid}$ to optimize the model parameters. Since we did not employ a discriminator for adversarial training, no adversarial loss was used. We used a weighted sum of $L_1$ (mean absolute error), $L_2$ (mean square error) and $L_{LPIPS}$ as shown below:

$$L_{hybrid} = (1 - \alpha)L_1 + \alpha L_2 + L_{LPIPS} \tag{10}$$

Due to limited computational resources, the *maximum* number of training epochs was set to 300 for CelebA-HQ and 50 for ImageNet experiments. All experiments were run on a single 80 GB Nvidia A100 GPU. We used AdamW optimizer ($\alpha = 0.001, \beta_1 = 0.9, \beta_2 = 0.999, \epsilon = 10^{-8}$) with a weight decay of 0.01 during initial epochs and then used stochastic gradient descent (SGD) with a learning rate of 0.001 and momentum $= 0.9$ during the final epochs (Keskar & Socher, 2017; Jeevan & sethi, 2022). We used the maximum batch-size that could be accommodated in a single GPU for our experiments. We used an embedding dimension ($C$) of 128 in all the Wave modules. Each Wave module has 4 WaveMix blocks unless otherwise specified.

### 4.3 Quantitative results

We compared our models with the other state-of-the-art baselines on the $256 \times 256$ CelebA-HQ dataset for narrow, medium, and wide masks, as shown in Table 1. WavePaint consistently outperforms all other models on different mask configurations in FID. Even though we have reported LPIPS in the table, FID a better metric because the pixel-wise evaluation of LPIPS punishes diverse inpainting systems for large

Table 1: Quantitative comparison of reconstruction quality of WavePaint (which is parameter-efficient and does not require adversarial training) with other methods (which use adversarial training to improve results) on CelebA-HQ dataset masked with narrow, medium, and wide masks (as done in LaMa (Suvorov et al., 2021a)) using two metrics – Learned perceptual image patch similarity (LPIPS) and Fréchet inception distance (FID) – with the best WavePaint results highlighted in bold and the results of other models which performed better than WavePaint in red.

| | | **CELEBA-HQ** $(256 \times 256)$ | | | | | |
|---|---|---|---|---|---|---|---|
| **MODEL** | **#PARAM.↓** | **NARROW MASKS** | | **MEDIUM MASKS** | | **WIDE MASKS** | |
| | | **FID↓** | **LPIPS↓** | **FID↓** | **LPIPS↓** | **FID↓** | **LPIPS↓** |
| CoModGAN (Zhao et al., 2021) | 109 M | 16.8 | 0.079 | 19.4 | 0.092 | 24.4 | 0.102 |
| AOT GAN (Zeng et al., 2021) | 15 M | 6.67 | 0.081 | 7.28 | 0.089 | 10.3 | 0.118 |
| RegionWise (Ma et al., 2019) | 47 M | 11.1 | 0.124 | 7.52 | 0.101 | 8.54 | 0.121 |
| DeepFill v2 (Yu et al., 2019) | 4 M | 12.5 | 0.130 | 9.05 | 0.105 | 11.2 | 0.126 |
| EdgeConnect (Nazeri et al., 2019b) | 22 M | 9.61 | 0.099 | 7.56 | 0.095 | 9.02 | 0.120 |
| LaMa-Fourier (Suvorov et al., 2021a) | 27 M | 7.26 | 0.085 | 6.13 | 0.080 | 6.96 | 0.098 |
| WavePaint | 3 M | 8.03 | 0.115 | 8.87 | 0.123 | 21.3 | 0.155 |
| WavePaint | 12 M | **5.34** | 0.082 | 5.63 | 0.088 | 7.27 | 0.111 |
| WavePaint | 13 M | 5.43 | **0.081** | **5.47** | **0.086** | **6.84** | **0.107** |

Table 2: Quantitative comparison of reconstruction quality of WavePaint with other models for inpainting narrow masks in CelebA-HQ dataset for image resolution of $512 \times 512$ (better results in bold).

| | **CELEBA-HQ** $(512 \times 512)$ | |
|---|---|---|
| **MODELS** | **#PARAM.** | **NARROW MASK FID ↓** |
| LaMa (Suvorov et al., 2021a) | 27 M | 4.05 |
| ICT (Wan et al., 2021) | 150 M | 6.28 |
| MADF (Zhu et al., 2021) | 85 M | 3.39 |
| AOT GAN (Zeng et al., 2021) | 15 M | 4.65 |
| EdgeConnect (Nazeri et al., 2019b) | 22 M | 10.58 |
| **WavePaint** | **10 M** | **3.36** |

holes (Li et al., 2022). It also has to be noted all other models have much larger parameter count and employ adversarial training using a discriminator. Since WavePaint does not employ a discriminator ,it is light-wight, and it can be trained faster than GANs and diffusion models. Similarly, we see from Table 2 that WavePaint can outperform all other models on narrow mask inpainting of $512 \times 512$ CelebA-HQ dataset.

Among models that use 2D-DWT for inpainting, WaveFill (Yu et al., 2021) uses self-attention with adversarial training and performs better than others. When compared to WaveFill (about 50 M parameters), WavePaint

Table 3: Multi-aspect comparison of WavePaint with LaMa (Suvorov et al., 2021a) on parameters, resource-consumption and speed for CelebA-HQ dataset with narrow masks on a single 24 GB RTX 3090 GPU with a batch size of 10 (better results in bold).

| **MODEL** | **#PARAM** | **FID↓** | **GPU** | **THROUGHPUT (FPS)** | |
|---|---|---|---|---|---|
| | | | | **INFERENCE** | **TRAIN** |
| LaMa (Suvorov et al., 2021a) | 27 M | 7.26 | 23 GB | 32 | 11 |
| WavePaint | **5 M** | **7.09** | **11 GB** | **105** | **32** |

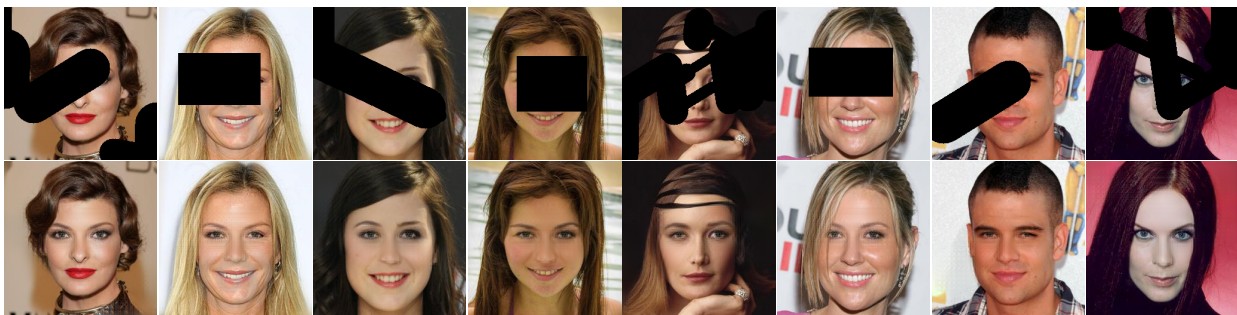

Figure 5: Wide-masked images from the CelebA-HQ dataset (top row) and their inpainted versions generated by WavePaint (bottom row). More qualitative results are provided in Appendix.

is lighter (less than one-third parameters) and outperforms WaveFill in terms of FID in $256 \times 256$ CelebA-HQ dataset with higher inference speed ($3\times$ faster).

Since LaMa (Suvorov et al., 2021a) was the most resource-efficient model for inpainting which was also outperforming the other models, we only compared WavePaint with LaMa in Table 3 to analyse its resource-efficiency. We see that WavePaint requires less than one-fifth of the parameters of LaMa to outperform it in FID metric. WavaPaint is also $\sim 3\times$ faster than LaMa in both inference and training speeds and utilizes less than half the GPU RAM required by LaMa. The actual speed of WavePaint will be even higher (almost twice) as we used same batch size of 10 for both LaMa and WavePaint in out experiments. Our results clearly shows that WavePaint is more resource and parameter-efficient than LaMa. The high resource-efficiency of WavePaint can be attributed to the resource-efficient token-mixing using WaveMix blocks which processes the image at a lower resolution due to lossless downsampling property of 2D-DWT.

### 4.3.1 Comparision with Diffusion Models

We could not compare WavePaint with latest diffusion models such as RePaint (Lugmayr et al., 2022) because diffusion is a much slower process of image generation and we were constrained by the available computational resources. We tried RePaint with 1000 diffusion steps and 10 times resampling, but the model had over 500 M parameters and inference on each image took around 100s. RePaint (Lugmayr et al., 2022) had reported that quantitative results of LaMa (Suvorov et al., 2021a) are better than that of RePaint in wide and narrow mask inpainting on ImageNet and CelebA-HQ datasets. Since, the quantitave performance of WavePaint is better than LaMa, we can claim that it is also better than RePaint.

### 4.4 Qualitative Results

The images generated by WavePaint on ImageNet dataset are shown in Figure 4. We can see that WavePaint completes textures and missing details by completing the lines and filling in details. The images generated by WavePaint on CelebA-HQ dataset are shown in Figure 6. WavePaint can fill in missing details of facial features, color, texture, eyes and eyebrows even if major parts of the image are masked. We show qualitative comparisons with LaMa in Figure 9. We observe that WavePaint can fill in details like eyes and eyebrows better than LaMa. Analysing the images inpainted by WavePaint, we observed presence of fine-texture artifacts in inpainted portions in some of the images when zoomed in. These artifacts were of similar color and texture of the background, hence almost invisible in most cases. More examples of qualitative results with different mask categories are provided in the Appendix.

## 5 Ablation Studies

Multiple ablation experiments were conducted to optimize the network hyper-parameters and understand the utility of the network components. The quantitative performance of WavePaint using different hyper-parameters on CelebA-HQ and ImageNet datasets are shown in Table 4. Table 5 shows the performance of

Table 4: Comparison of image reconstruction and throughput metrics for WavePaint architectures of different sizes that use level-1 2D-DWT and 4 WaveMix blocks per modules by varying the number of modules on 2,000 images of CelebA-HQ (as used by LaMa (Suvorov et al., 2021a), 36,500 images of Places-365 standard validation set and 50,000 images of ImageNet validation set on a single 80 GB A100 GPU.

| MODEL | #MODULES | DEPTHCONV | PARAMS | NARROW MASKS | | MEDIUM MASKS | | WIDE MASKS | | THROUGHPUT (FPS) | |
|---|---|---|---|---|---|---|---|---|---|---|---|
| | | | | FID↓ | LPIPS↓ | FID↓ | LPIPS↓ | FID↓ | LPIPS↓ | INFERENCE | TRAIN |
| | | | | | | CelebA-HQ ($256 \times 256$) | | | | | |
| WavePaint | 2 | No | 3.3 M | 11.1 | 0.148 | 13.9 | 0.148 | 33.7 | 0.176 | 356 | 165 |
| WavePaint | 2 | Yes | 3.3 M | 8.03 | 0.115 | 8.87 | 0.123 | 21.3 | 0.155 | 322 | 145 |
| WavePaint | 3 | Yes | 5.0 M | 7.09 | 0.103 | 6.96 | 0.104 | 10.2 | 0.131 | 275 | 99 |
| WavePaint | 5 | Yes | 8.4 M | 6.56 | 0.096 | 6.62 | 0.098 | 8.83 | 0.122 | 167 | 60 |
| WavePaint | 6 | Yes | 10 M | 5.53 | 0.085 | 5.59 | 0.090 | 7.22 | 0.112 | 133 | 50 |
| WavePaint | 7 | Yes | 12 M | **5.34** | 0.082 | 5.63 | 0.088 | 7.27 | 0.111 | 117 | 42 |
| WavePaint | 8 | Yes | 13 M | 5.43 | **0.081** | **5.47** | **0.086** | **6.84** | **0.107** | 105 | 37 |
| | | | | | | Places-365 ($256 \times 256$) | | | | | |
| WavePaint | 8 | Yes | 13 M | 4.63 | 0.152 | 4.82 | 0.127 | 9.74 | 0.157 | 97 | 35 |
| | | | | | | ImageNet ($224 \times 224$) | | | | | |
| WavePaint | 2 | Yes | 3.3 M | 3.26 | 0.134 | 3.72 | 0.108 | - | - | 333 | 213 |
| WavePaint | 3 | Yes | 5.0 M | 3.21 | 0.138 | 3.47 | 0.106 | - | - | 305 | 126 |

Table 5: Comparison of performance of WavePaint architecutres with different levels of 2D-DWT using three modules and four WaveMix blocks per module on CelebA-HQ dataset show improved performance due to the rapid expansion of receptive fields while using multi-level 2D-DWT token-mixing

| MODEL | PARAM | NARROW MASKS | | MEDIUM MASKS | | WIDE MASKS | | THROUGHPUT (FPS) | |
|---|---|---|---|---|---|---|---|---|---|
| | | FID↓ | LPIPS↓ | FID↓ | LPIPS↓ | FID↓ | LPIPS↓ | INFERENCE | TRAIN |
| Level 1 | 5.0 M | **7.09** | 0.103 | **6.96** | 0.104 | 10.2 | 0.131 | 275 | 99 |
| Level 2 | 7.6 M | 7.12 | 0.095 | 7.16 | 0.096 | **9.16** | 0.119 | 222 | 78 |
| Level 3 | 10 M | 7.74 | **0.094** | 7.62 | **0.092** | 9.26 | **0.112** | 200 | 67 |

Table 6: Performance of WavePaint with 8 WaveMix blocks by varying the number of modules on a subset of ImageNet dataset

| #MODULES | #WAVEMIX BLOCKS | #PARAM | LPIPS |
|---|---|---|---|
| 1 | 8 | 3.0 M | 0.085 |
| 2 | 4 | 3.3 M | 0.079 |
| 4 | 2 | 4.0 M | 0.079 |

WavePaint which uses WaveMix blocks with multi-level 2D-DWT. Using higher levels of DWT can improve the performance of the model due to the exponential increase in receptive field. WavePaint can also generalise to larger mask sizes even from training on smaller masks. The generalization results by training using medium mask and infering on wide masks are provided in Appendix.

Table 6 shows the performance of WavePaint with 8 WaveMix blocks arranged in different number of modules. Results shows that having less number of modules with large number of WaveMix blocks is more parameter-efficient but results in poor performance. When we decrease the number of WaveMix blocks in each module and increase the number of modules, the model become larger with higher parameter count. Modules with four Waveblocks each retain parameter-efficiency without degrading performance.

Removing DepthConv block from WavePaint reduces the FID score by 38% and increases the training and inference throughput by 14%. Since, depth-wise convolution is a highly parameter efficient operation, its removal only reduces the number of parameters by less than 1%. Therefore, adding DepthConv block in each module is beneficial for the network as it aids the WaveMix block with further spatial token-mixing.

## 6 Conclusion and Future Work

This paper proposes using multi-level 2D-DWT token-mixing for the less-explored task of image inpainting. The performance of the proposed model is comparable to much larger models and those that use adversarial training on CelebA-HQ dataset. Moreover, our model uses only a fraction of the parameters, consumes less GPU RAM and is multiple times faster in training and inference compared to other models such as LaMa (Suvorov et al., 2021a). The faster receptive field expansion leading to availability of global context information can help these models do image reconstruction.

A possible direction of future work is to develop resource-efficient image generation models using WavePaint trained in an adversarial or diffusion setting. This is likely to further suppress the fine-texture artifacts that are barely visible in some of the image regions generated by WavePaint. Overall, this paper highlights the potential of using token-mixing that exploit natural image priors as an alternative to vision transformers and CNNs for resource-efficient image inpainting.

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

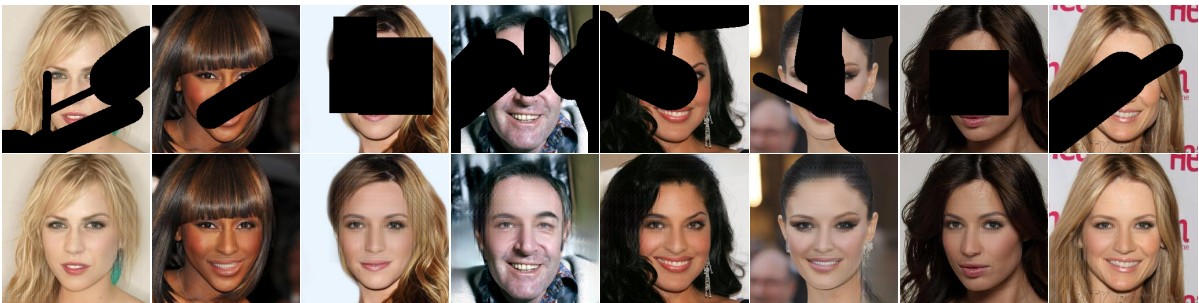

Figure 6: Wide-masked images from the CelebA-HQ dataset (top row) and their inpainted versions generated by WavePaint (bottom row) Inpainted images (bottom row)

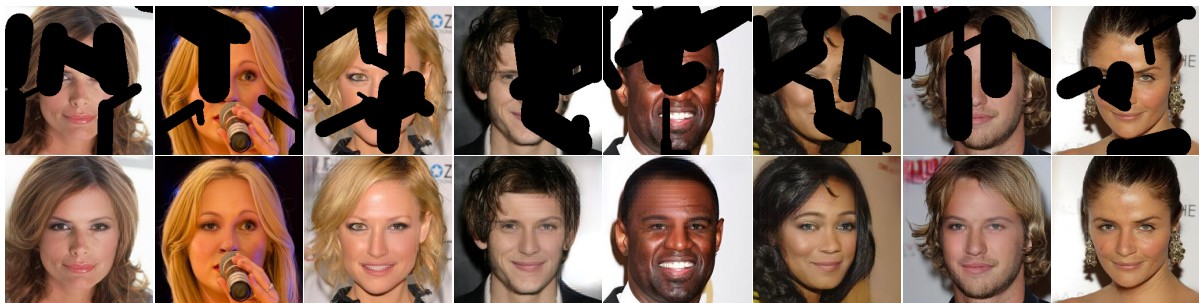

Figure 7: Medium-masked images from the CelebA-HQ dataset (top row) and their inpainted versions generated by WavePaint (bottom row) Inpainted images (bottom row)

# A    Appendix

## A.1    Qualitative Results

The qualitative results of inpainting by WavePaint for wide, medium, and narrow masks are shown in Figure 6, 7, and 8, respectively. All the images were generated by WavePaint (12M parameters) trained only on wide masks. We see that WavePaint is able to fill in missing region with facial features. It is able to match the eye colours, eyebrows etc from the visible region to the generated parts.

In some images we observe the presence of fine-texture artifacts near the masked portions. We ran some experiments by training WavePaint in adversarial settings using a discriminator and observed that these artifacts disappear. For future work, we will create an adversarially trained WavaPaint model which would produce more realistic images and will be resource-efficient.

Additional qualitative comparison of images inpainted using WavePaint with those inpainted using LaMa (Suvorov et al., 2021a) are shown in Figure 9.

## A.2    WaveMix Block (Jeevan et al., 2023)

WaveMix blocks with one and three levels of 2D-discrete wavelet transform (2D-DWT) are shown in Figure 10 and 11 respectively. WaveMix block having a single level of 2D-DWT is called WaveMix-Lite. Denoting input and output tensors of the WaveMix block by $\mathbf{x}_{in}$ and $\mathbf{x}_{out}$, respectively; level of the wavelet transform by $l \in \{1...L\}$, the four wavelet filters along with their downsampling operations at each level by $w_{aa}^l, w_{ad}^l, w_{da}^l, w_{dd}^l$ ($a$ for approximation, $d$ for detail); convolution, multi-layer perceptron (MLP), transposed convolution (upconvolution), and batch normalization operations by $c$, $m$, $t$, and $b$, respectively; and their respective trainable parameter sets by $\xi$, $\theta_l$, $\phi_l$, and $\gamma_l$, respectively; concatenation along the channel dimension by $\oplus$, and point-wise addition by $+$, the operations inside a WaveMix block can be expressed using the following equations:

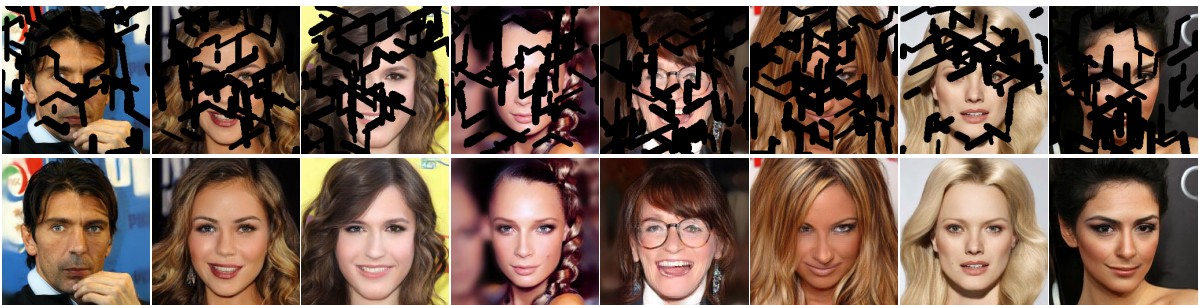

Figure 8: Narrow-masked images from the CelebA-HQ dataset (top row) and their inpainted versions generated by WavePaint (bottom row) Inpainted images (bottom row)

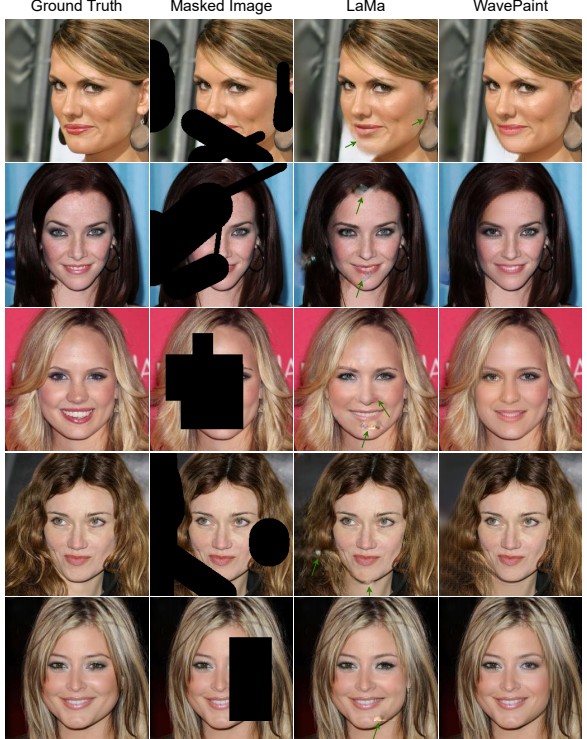

Figure 9: Qualitative comparison of inpainted images generated by WavePaint and LaMa (Suvorov et al., 2021a). The green arrows point to improper image completion.

$$\mathbf{x}_0 = c(\mathbf{x}_{in}, \xi); \mathbf{x}_{in} \in \mathbb{R}^{H \times W \times C}, \mathbf{x}_0 \in \mathbb{R}^{H \times W \times C/4} \tag{11}$$

$$\mathbf{x}_l = [w_{aa}^l(\mathbf{x}_0) \oplus w_{ad}^l(\mathbf{x}_0) \oplus w_{da}^l(\mathbf{x}_0) \oplus w_{dd}^l(\mathbf{x}_0)]; \mathbf{x}_l \in \mathbb{R}^{H/2^l \times W/2^l \times 4C/4}, l \in \{1...L\} \tag{12}$$

$$\hat{\mathbf{x}}_l = [\mathbf{x}_l \oplus \tilde{\mathbf{x}}_{l+1}], \quad \hat{\mathbf{x}}_L = \mathbf{x}_L; \quad l \in \{1...L-1\} \tag{13}$$

$$\tilde{\mathbf{x}}_l = b(t(m(\hat{\mathbf{x}}_l, \theta_l), \phi_l), \gamma_l); \quad \tilde{\mathbf{x}}_l \in \mathbb{R}^{H/2^{l-1} \times W/2^{l-1} \times C_l} \forall l > 1 \quad C_l = C/2, \quad C_1 = C, \quad l \in \{1...L\} \tag{14}$$

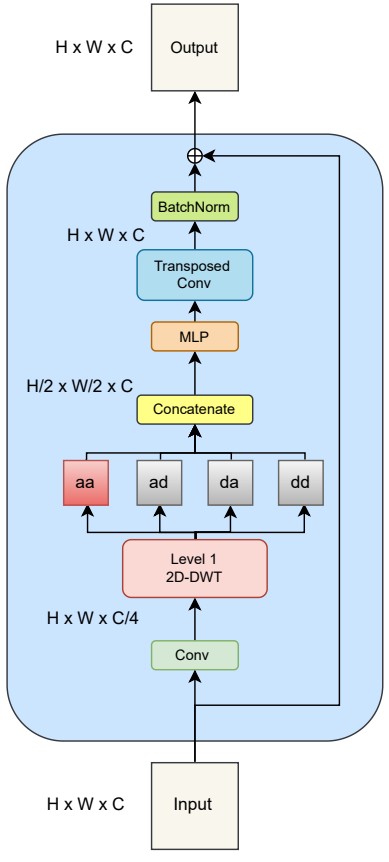

WaveMix-Lite Block (1-level 2D-DWT)

Figure 10: WaveMix block architecture using level-1 2D-discrete wavelet transform. The image is take from (Jeevan et al., 2023)

Table 7: Mask generalization performance of WavePaint when using medium masks for training.

| | | CELEBA-HQ ($256 \times 256$) | | | | | |
|---|---|---|---|---|---|---|---|
| | | NARROW MASKS | | MEDIUM MASKS | | WIDE MASKS | |
| MODEL | #PARAM↓ | FID↓ | LPIPS↓ | FID↓ | LPIPS↓ | FID↓ | LPIPS↓ |
| WavePaint | 12 M | 4.68 | 0.073 | 4.98 | 0.082 | 6.69 | 0.108 |
| WavePaint | 13 M | 4.44 | 0.070 | 4.79 | 0.080 | 6.27 | 0.104 |

$$\mathbf{x}_{out} = \tilde{\mathbf{x}}_1 + \mathbf{x}_{in}; \quad \mathbf{x}_{out} \in \mathbb{R}^{H \times W \times C} \tag{15}$$

## B  Mask Generalization

In all the reported results, WavePaint was trained only on wide masks and tested on narrrow, medium and wide masks. For checking the generalisation ability of WavePaint to larger mask sizes, we trained WavePaint on medium masks and tested its performance on wide masks. The results are shown in Table 7. We see that WavePaint is providing good performance in wide masks even when trained of medium masks. This shows that WavePaint is able to generalize well on larger unseen masks during inference.

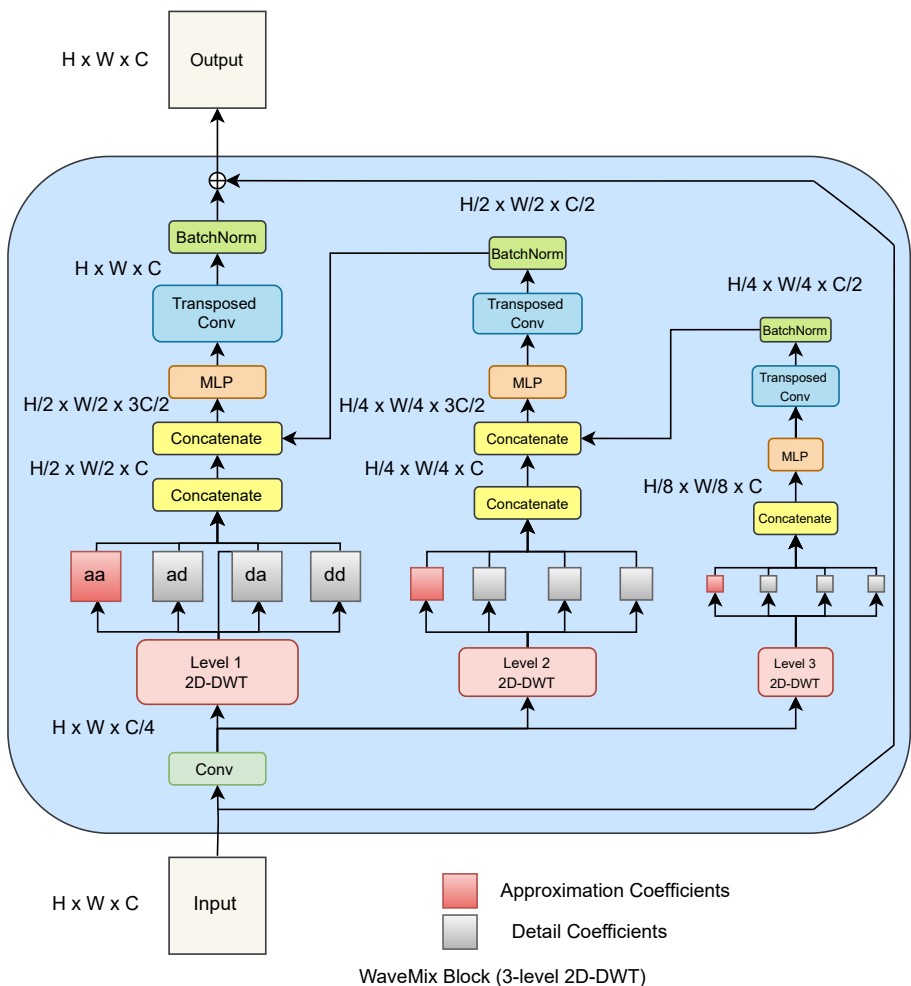

Figure 11: Details of the WaveMix block with 3 levels of 2D-DWT. The image is take from (Jeevan et al., 2023)

