# OpenReview forum: "WavePaint: Resource-efficient Token-mixer for Self-supervised Inpainting"
_TMLR — Rejected by TMLR_

### Review · Reviewer_ofLk · 2024-11-11

**Summary Of Contributions:**

The paper focuses on the problem of image inpainting and restoration using resource-efficient architectures. Authors propose an architecture that performs spatial and multi-resolution token-mixing within a fully-convolutional model by using 2D discrete wavelet transforms. Experiments are performed on CelebA-HQ, Places-365 and ImageNet datasets, demonstrating a both quantitatively superior, and computationally more efficient solution to the problem of image inpainting.

**Audience:**

Yes

**Claims And Evidence:**

Yes

**Requested Changes:**

- There are no comparisons to diffusion models implemented for similar inpainting tasks. Although they are clearly not of the same scale in terms of the computational overhead or number of model parameters, at least some simple DDPM [Ho et al 2020], Repaint [Lugmayr et al. 2022] or DDRM [Kawar et al. 2022] type baselines can be included for comparisons to an actual state-of-the-art in this domain as a quantitative reference.

- Please include the exact narrow/medium/wide mask sizes (in terms of pixels) also in each table's caption as a clearer reference.

Minor:

- The tables look a bit odd in appearance and organization with the current LateX usage. Perhaps it can be improved using the general text/table editing guidelines.

- The equations in pages 5 & 6 can be somewhat shortened in space for better presentation and writing.

**Strengths And Weaknesses:**

Strengths:
- Authors propose a novel and computationally efficient solution that is engineered for the problem of image inpainting.
- The scale of the experiments in terms of the datasets and considered resolutions are thorough.

Weaknesses:
- The scope is limited to the image inpainting application.
- More recent methodological solutions (e.g., based on generative diffusion models) are not included in the experiments.

---

> ### Author Response · Authors · 2024-11-23
> **Response to Reviewer ofLk**
>
> We extend our heartfelt thanks for your valuable feedback and insightful comments, which have significantly enhanced the quality and clarity of our work. We deeply appreciate the time and effort you devoted to reviewing our manuscript.
>
> The primary objective of this research was to develop an efficient model with faster inference compared to the previous state-of-the-art, which are mostly GAN-based inpainting models such as LaMa. We did not include comparisons with diffusion models, as they require hundreds of iterative steps, resulting in extremely slow inference. In Section 4.3 (page 8, first paragraph), we referenced a diffusion-based model, RePaint, which was specifically designed for image inpainting. RePaint's results, as reported in its study, included comparisons with LaMa, showing that LaMa achieved superior quantitative performance (LPIPS) for both narrow and wide masks. Due to the computational resource constraints associated with RePaint—being a large model with over 500 million parameters and requiring approximately 100 seconds per image for inference—we were unable to perform experiments with it. However, since our model demonstrates better quantitative performance than LaMa across narrow, medium, and wide masks, we can reasonably infer that our model also outperforms RePaint, addressing both efficiency and effectiveness.
>
> The exact sizes of the narrow, medium, and wide masks (in terms of pixels) were determined using the same mask generation strategy described in the LaMa paper. The masked area, representing the average number of masked pixels per image, is restricted to less than 50% across all test images for narrow, medium, and wide masks. All mask types cover between 10-50% of the total image pixels. The primary difference lies in the distribution of masks within the 40-50% range, with a higher proportion of images falling in this range for wide masks. Additionally, the key distinction among narrow, medium, and wide masks is the thickness of the masks, which progressively increases from narrow to wide. This distinction is further detailed and visualized in the supplementary materials of the LaMa paper (page 6).
>
> We will improve the tables in accordance with the general text/table editing guidelines.
> We will also shorten the equations in the revision.

---

### Review · Reviewer_kVFJ · 2024-11-18

**Summary Of Contributions:**

This paper presents a method for image inpainting. The key idea is to adopt a module that applies discrete wavelet transform to the image (feature maps), followed by a MLP for feature extraction and image transform. The authors claim that this approach allows the model to have larger receptive field with a shallower architecture, hence the model is easier to train (without a diffusion process and adversarial loss), while performing as good as other state-of-the-art methods that are based on GANs. Specifically, the authors claim that using wavelet transform yields better results compared to LaMa which adopts the Fourier transform for the same purpose, even without the adversarial loss.

**Audience:**

Yes

**Broader Impact Concerns:**

I have no concern on the ethical implications of this work.

**Claims And Evidence:**

No

**Requested Changes:**

Given the weaknesses I summarized above, I propose the following changes:
1. Add a discussion regarding the potential problems in the results and discuss why the proposed method still has advantages over existing approaches.
2. Explain the methods in better details with more analysis on how the proposed method works.

**Strengths And Weaknesses:**

# Strength
The idea of using DWT in neural network is interesting, despite that it has already been used in other image restoration tasks in existing works.

# Weakness
1. [**Results**] My first major concern is on the results. As we can clearly see from all shown figures, the images produced by the proposed method suffer severe checkerboard artifacts. To name one of the example, in Fig. 9 row 3, the checkerboard artifacts are so significant that the lady's teeth are all messed up. Row 4 in the same figure have very significant artifacts in the area of hairs. Given these results, I can hardly agree that the proposed method performs better than LaMa. Hence, I cannot agree that the authors' claims are well supported by the experiments. Besides, the authors claim that the proposed method has benefits in complexity. This is not supported by the results either, as from table 1 I don't see a significant improvement in complexity.

2. [**Technical Soundness**] If I understand correctly, the authors only apply a 1-layer DWT in each of the wavelet module (as shown in Fig. 3). With a Haar wavelet, the 1-layer transform will only look at the one adjacent neighbor in each direction (equivalent to 2x2 kernels). In this case, I don't see why the authors claim that the transform can significantly enlarge  the receptive field.

3. [**Treatment of content**] The descriptions of the method in the manuscript is not clear nor self-contained. As I understand, the wavelet transform is the important part. Although I think people can guess the meaning of "aa", "ad" etc, the authors didn't clearly define these terms. At the same time, a lot of space is wasted for Eq. (1) - (9) which provide no information in addition to Fig. 3, as well as large margin in page 10. The authors should spend more space analyzing the effect of the DWT, in addition to a big table just showing the final results with different settings, to help readers understand why the proposed method can work.

---

> ### Author Response · Authors · 2024-11-23
> **Response to Reviewer kVFJ**
>
> We sincerely thank you for your invaluable feedback and thoughtful comments, which have greatly improved the quality and clarity of our work. We deeply appreciate the time and effort you dedicated to reviewing our manuscript.
>
> While our qualitative results occasionally exhibit checkerboard artifacts, the quantitative metrics, such as FID, consistently demonstrate superior performance. These artifacts stem from the use of transposed convolution layers, which can be mitigated by replacing them with pixel shuffle or bilinear upsampling techniques. As shown in Table 3, our method achieves lower computations and parameter count compared to LaMa, further highlighting its efficiency.
> In our approach, multi-level wavelet transforms can be incorporated into each WaveMix block, exponentially expanding the receptive field. Additionally, the use of large-kernel depthwise convolutions complements this by rapidly increasing the receptive field alongside the transposed convolutions employed for upsampling.
> In our revision, we will provide clearer explanations of our terminology and conduct a more detailed analysis of the wavelet transform's impact. Furthermore, we will include a discussion elaborating on the advantages of our proposed method over existing approaches.

---

### Review · Reviewer_oZhj · 2024-11-21

**Summary Of Contributions:**

This paper proposes an in-painting setup using WaveMix [Jeevan et al]. The idea is to replace convolutions (con) or vision transformer (vit) backbones with a WaveMix backbone. The claim is that (from Jeevan et al.) WaveMix, which consists of blocks containing wavelet transforms instead of convs or vits, is computationally cheaper, while offering the same performance. In inpainting setups, the authors point out that their method does away with the need for adversarial fine tuning.

In this work, the problem of interest is in-painting - a generative problem. The input consists of images concatenated with a mask (generated for training), and the network learns to reconstruct the unmasked (i.e original) image as output. As stated, wavemix blocks are used in the architecture, with convs, DWT and transposed convs contained within. The setup also has depth wise convolutions as additional processing modules. The network is trained with a combination of L1, L2 and LPIPS losses.

Results are shown for the CelebA dataset to measure FID and LPIPS metrics, tested with narrow, medium and wide masks. They seem competitive with provided baselines. They also show speed metrics (FPS) for training and inference and argue that their method fares well in that regard.

**Audience:**

Yes

**Claims And Evidence:**

Yes

**Requested Changes:**

Can the authors shed more light on what the depth wise convolution block does in this work?
I would also appreciate (as stated above) some treatment of diffusion models. I do not see any of the competitive methods being labeled as diffusion models.

**Strengths And Weaknesses:**

Strengths:
+ Conceptually simple method - replace blocks with WaveMix blocks - and we have a decent inpainter
+ Training appears to be straightforward in that it is not plagued by problems like instabilities in GANs
+ Results, as shown are attractive. Quality is good and speed is much better than baseline (Lama: Suvorov et al.)

Weaknesses:
- Novelty: The method appears to be a straightforward application of the WaveMix setup to the inpainting problem. While this is not as such a weakness, I am left wondering if it follows simply from the original paper by Jeevan et al. (which had covered segmentation). Nonetheless, I think it is a good extension, and potentially useful.
- Technical contribution is limited to change in 'application'.
- I would have liked to have seen some treatment of diffusion models in this work, as it is the SOTA family of methods generally in image generation problems.

---

> ### Author Response · Authors · 2024-11-23
> **Response to Reviewer oZhj**
>
> We extend our heartfelt thanks for your valuable feedback and insightful comments, which have significantly enhanced the quality and clarity of our work. We deeply appreciate the time and effort you devoted to reviewing our manuscript.
>
> Depthwise convolution serves as an additional token-mixing operation, complementing the wavelet-based token-mixing process. While wavelet token-mixing is a parameter-free operation, the model learns primarily through parameters in MLPs and transposed convolution blocks following the wavelet transforms in each layer. This leads to having to add more WaveMix layers for increasing expressiveness. By incorporating depthwise convolution, we enhance the network's expressiveness with minimal increase in parameter count, as depthwise convolution is inherently more parameter-efficient than standard convolution. Our experiments empirically demonstrate that the inclusion of depthwise convolution markedly improves WavePaint's performance without significantly increasing parameter count or training time.
>
> The primary objective of this research was to develop an efficient model with faster inference compared to the previous state-of-the-art, which are mostly GAN-based inpainting models such as LaMa. We did not include comparisons with diffusion models, as they require hundreds of iterative steps, resulting in extremely slow inference. In Section 4.3 (page 8, first paragraph), we referenced a diffusion-based model, RePaint, which was specifically designed for image inpainting. RePaint's results, as reported in its study, included comparisons with LaMa, showing that LaMa achieved superior quantitative performance (LPIPS) for both narrow and wide masks.  Due to the computational resource constraints associated with RePaint—being a large model with over 500 million parameters and requiring approximately 100 seconds per image for inference—we were unable to perform experiments with it. However, since our model demonstrates better quantitative performance than LaMa across narrow, medium, and wide masks, we can reasonably infer that our model also outperforms RePaint, addressing both efficiency and effectiveness.

---

### Decision · Action_Editor_87ey · 2025-01-05

**Recommendation:** Reject

**Comment:**

The AC has carefully reviewed the paper and agree with the majority of reviewers' recommendations for rejection. In its current form, the experimental results are unconvincing, with severe checkerboard artifacts in the generated images, which undermine the claims of superior performance over LaMa. Furthermore, the use of a 1-layer DWT with Haar wavelets does not substantiate the claimed significant enlargement of the receptive field, raising concerns about the technical validity of the method.

**Audience:**

The researchers who focus on image restoration may be interested.

**Claims And Evidence:**

This paper has significant limitations and, therefore, cannot be accepted.

First, the experimental results are unconvincing. The visual quality of the results can not demonstrate the advantages of the proposed method. The reviewers noted that the visual quality is "far from acceptable", and the AC agrees that addressing this issue requires fundamental design improvements rather than simple adjustments.

Second, the method design is also unconvincing. In particular, the rationale behind the 1-layer wavelet transform is problematic, and this issue remains unresolved even after the rebuttal.

Additionally, the paper lacks comparisons with recent state-of-the-art methods, as most of the comparisons are conducted against methods proposed around 2021.

In conclusion, the claims supporting the paper, including the advantages of the proposed method in terms of generalization and effectiveness, are flawed and raise significant concerns.

**Resubmission Of Major Revision:**

The authors may consider submitting a major revision at a later time.